# Clinical practice guidelines for acute otitis media in children: a systematic review and appraisal of European national guidelines

Hijiri G Suzuki [iD],[1] Juan Emmanuel Dewez [iD],[1] Ruud G Nijman [iD],[2] Shunmay Yeung [iD][1]

¹Department of Clinical Research, Faculty of Infectious and Tropical Disease, London School of Hygiene and Tropical Medicine, London, UK
²Faculty of Medicine, Department of Infectious Diseases, Section of Paediatric Infectious Diseases, Imperial College London, London, UK

**Correspondence to**
Dr Shunmay Yeung;
shunmay.yeung@lshtm.ac.uk

## ABSTRACT

**Objectives** To appraise European guidelines for acute otitis media (AOM) in children, including methodological quality, level of evidence (LoE), astrength of recommendations (SoR), and consideration of antibiotic stewardship.

**Design** Systematic review of the literature.

**Data sources** Three-pronged search of (1) databases: Medline, Embase, Cochrane library, Guidelines International Network and Trip Medical Database; (2) websites of European national paediatric associations and (3) contact of European experts. Data were collected between January 2017 and February 2018.

**Eligibility criteria** National guidelines of European countries for the clinical management of AOM in children aged <16 years.

**Data extraction and synthesis** Data were extracted using tables constructed by the research team. Guidelines were graded using AGREE II criteria. LoE and SoR were compared. Guidelines were assessed for principles of antibiotic stewardship.

**Results** AOM guidelines were obtained from 17 or the 32 countries in the European Union or European Free Trade Area. The mean AGREE II score was ≤41% across most domains. Diagnosis of AOM was based on similar signs and symptoms. The most common indication for antibiotics was tympanic membrane perforation/otorrhoea (14/15; 93%). The majority (15/17; 88%) recommended a watchful waiting approach to antibiotics. Amoxicillin was the most common first-line antibiotic (14/17; 82%). Recommended treatment duration varied from 5 to 10 days. Seven countries advocated high-dose (75–90 mg/kg/day) and five low-dose (30–60 mg/kg/day) amoxicillin. Less than 60% of guidelines used a national or international scale system to rate level of evidence to support recommendations. Under half of the guidelines (7/17; 41%) referred to country-specific microbiological and antibiotic resistance data.

**Conclusions** Guidelines for managing AOM were similar across European countries. Guideline quality was mostly weak, and it often did not refer to country-specific antibiotic resistance patterns. Coordinating efforts to produce a core guideline which can then be adapted by each country may help improve overall quality and contribute to tackling antibiotic resistance.

### Strengths and limitations of this study

► The methodology includes the use of a comprehensive three-pronged search strategy with no language restrictions to identify guidelines from across Europe, the use of a standardised and internationally recognised guideline appraisal tool (AGREE II), the assessment of levels of evidence and strength of recommendations and the assessment of whether antibiotic stewardship, a key measure to reduce antimicrobial resistance (AMR), was considered.

► The review focused only on AOM without complications; guidelines for complex otitis media requiring specialist otolaryngology input were not included. Another limitation is the consideration of whether guidelines developers used country-specific AMR patterns to assess if the recommendations of antibiotics were based on AMR data. However, there is often wide heterogeneity in terms of AMR patterns within each country.

## INTRODUCTION

Acute otitis media (AOM) is one of the most common infections in childhood[1 2]; approximately 60% of children have had at least one episode by 4 years of age.[3] It is also one of the most frequently cited reasons for antibiotic prescription in children less than 3 years of age,[4 5] accounting for 14% of all antibiotic prescriptions in children in the UK.[6] While both bacterial and/or viral pathogens can cause AOM,[7 8] it is usually considered to be a bacterial complication of upper respiratory tract viral infection.[9]

The rationale for antibiotic prescription includes symptom control[10] and the prevention of rare but serious complications, including mastoiditis and meningitis.[11] However, studies show that up to 80% of cases resolve spontaneously without antibiotics,[12 13] and antibiotics are associated with the risk of side effects including vomiting, diarrhoea and rash.[13 14] In addition, the inappropriate

use of antibiotics has been identified as one of the key drivers of antibiotic resistance, a global health priority.[15–17] Emerging research has also demonstrated that longer antibiotic courses can lead to higher risks of resistance. Thus, providing clear guidance on appropriate antibiotic use in terms of the indications, choice and duration is considered important to help reduce antibiotic resistance.[18]

To promote antibiotic stewardship, the WHO recommends the development of treatment guidelines and the monitoring of local antibiotic resistance to inform the choice of antibiotics.[19] National guidelines for the first-line management of AOM may play a vital role in antibiotic stewardship.[20] To our knowledge, there has not been a systematic review of the quality and content of national guidelines for the management of AOM. The aim of this systematic review was to describe European guidelines for AOM in children to assess their methodological quality, to describe their evidence-based Strength of Recommendations (SoR) and to assess whether they incorporate consideration of antibiotic stewardship.

## METHODOLOGY

To ensure a comprehensive review of nationally endorsed guidelines, we used a three-pronged approach that included (1) a systematic database search; (2) a website search of European national societies and (3) expert consultation.

First, a systematic search of databases was carried out using Medline, Embase, Cochrane library, Guidelines International Network and Trip Medical Database from April 2017 to February 2018. Search terms were a combination of synonyms for (1) acute otitis media and (2) guidelines. Guidelines were included if they met the following eligibility criteria: (1) they were pertaining to the management of simple AOM, excluding the management of chronic or complex otitis media cases requiring specialist otolaryngology input; (2) they were national guidelines or endorsed by the national medical society from a European Union (EU) or European Free Trade Area (EFTA) country and (3) published from the year 2000 to present. The American Association of Pediatrics (AAP)[21] and the WHO[22] guidelines were also included for comparison as they are widely recognised and used internationally. The search included all European languages. An initial review of titles and abstracts was performed by one reviewer (HS). Additionally, the bibliographies of all guidelines were examined to identify further relevant resources (HS). Second, the websites of national paediatric associations listed by the European Paediatric Association/Union of National European Paediatric Societies and Associations were hand-searched (HS). Finally, a network of paediatric partners across Europe were contacted (RN, SY, JED and HS) to verify if the identified guidelines were the most up to date and widely used, and in cases where we had not managed to locate any guidelines, to assist in obtaining them. The choice of

search terms and final selection of full-text guidelines was performed by two reviewers (HS and JED) (see online supplementary files 1 and 2). If multiple national guidelines were found, the guideline judged to be most up to date, comprehensive and more commonly used in clinical practice was included after discussion between paediatrics partners and reviewers (HS and JED). Data were extracted using tables constructed by the research team.

### Patient and public involvement
This systematic review was performed without patient involvement.

### Guideline quality assessment
The AGREE II instrument was used independently by two reviewers (HS and JED) to determine the quality of each national guideline.[23] This is a standardised instrument that appraises the methodological framework of guideline development. The six domains assessed are (1) scope and purpose, (2) stakeholder involvement, (3) rigour of development including evidence base, (4) clarity of presentation, (5) applicability and (6) editorial independence. Domains were scored on a 1–7 scale; any score that varied by >3 out of 7 was discussed and revised if this was felt to be reasonable.

### Level of evidence and SoR
National scales for grading levels of evidence (LoE) and SoR were converted to Oxford Centre for Evidence Based Medicine (OCEBM) LoE and SoR (see online supplementary files 3 and 4). However, heterogeneity between grading systems meant that a meaningful comparison was difficult. Therefore in order to compare LoE between guidelines, we reviewed (1) whether guidelines used a national/international scale of evidence, (2) whether principles of risk versus harm were assessed, (3) whether strengths and limitations of evidence were assessed and (4) whether evidence was linked to a SoR. To allow for more meaningful comparison between guidelines, we used our scores for AGREE II items 11, 9 and 12 for the above (2), (3) and (4), respectively. We converted SoR into three categories: highest, moderate and lowest grade, indicated by shading of results in tables (tables 1 and 2).

### Antibiotic stewardship
As we were unable to find a standard scoring system to assess if a clinical guideline includes consideration of antibiotic stewardship, we based our methodology on a study by Elias *et al*.[24] We thus proposed six principles that demonstrate consideration of antibiotic stewardship based on the authors' consensus opinion. The principles are the inclusion in the guideline of (1) diagnostic criteria; (2) criteria for initiation of antibiotic therapy; (3) dosage; (4) route of administration; (5) what percentage of antibiotic recommendations was based on country-specific resistance patterns (ie, if two of three recommended antibiotics were supported by country-specific antibiotic resistance data, 67% was awarded) and (6) whether guidelines recommending amoxicillin or amoxicillin-clavulanic acid

**Table 1** Strength of Recommendations supporting immediate or watchful waiting approach to antibiotic administration in European, AAP and WHO guidelines

| Treatment approach | Strength of recommendation |
|---|---|
| **Immediate antibiotics for any AOM** | |
| WHO | Strong recommendation |
| **Immediate antibiotics for any AOM can be considered** | |
| Finland | A |
| USA | Recommendation |
| Czech Republic | No grade |
| **Watchful waiting approach** (except for indications outlined in table 2) | |
| France | A |
| Italy | A |
| Spain | A |
| Denmark | √ |
| Poland | B |
| Portugal | IIa |
| UK | B |
| Belgium | No grade |
| Germany | No grade |
| Ireland | No grade |
| Luxembourg | No grade |
| The Netherlands | No grade |
| Norway | No grade |
| Sweden | No grade |
| Switzerland | No grade |
| **Legend** | |
| Highest grade | |
| Moderate grade | |
| No grade | |

Note: There is no 'Lowest grade' in this table.
AAP, American Association of Pediatrics; AOM, acute otitis media; WHO, World Health Organisation.

based the dosage recommendation on country-specific resistance data. These two antibiotics were chosen because in contrast to other antibiotics, a higher dosage is recommended to overcome resistant strains.[25]

## RESULTS
### Overview of existing guidelines
The search retrieved 7340 records (figure 1). Of these, 19 guidelines were obtained. National guidelines were obtained from 17 of 32 European countries[26–42] (53%) (figure 2) and 2 non-European countries/organisations (USA and WHO). The majority of these were from Western Europe and Scandinavia. The intended audience of the obtained guidelines was mainly general

practitioners and paediatricians, although some included nurses and/or physician's assistants. Of note, 4 of 17 European guidelines clearly stated that they based their findings on other national guidelines, including those of the American Academy of Paediatrics, French Agence Française de Sécurité Sanitaire des Produits de Santé (now known as Agence Nationale de Sécurité du Médicament et des Produits de Santé) and UK Scottish Intercollegiate Guidelines Network (SIGN).

### Diagnostic criteria
Of note, 15 of 17 (88%) European guidelines outlined the signs and symptoms for diagnosing AOM (see online supplementary file 5) with considerable similarities between the guidelines. Twelve of 17 (71%) used strict combinations of three diagnostic criteria: (1) acute onset of symptoms (ie, otalgia, fever), (2) evidence of middle ear (ME) effusion (ie, tympanic membrane (TM) bulging of TM or otorrhoea on examination) and (3) inflammation of TM on examination.

### Otoscopy
Examination tools including standard otoscopy were advised by 15 of 17 (88%) European guidelines (see online supplementary file 6). Pneumatic otoscopy (9/15; 60%) and tympanometry (7/15; 50%) were also recommended.

### Additional investigations
No guidelines advised routine laboratory or radiographic investigations (see online supplementary file 7). Of note, 9 of 17 (53%) guidelines stated specific indications for carrying out investigations. Eight of 9 (89%) advised consideration of a culture sample of the ME via tympanocentesis, most commonly for treatment failure (6/9; 67%) and complications such as mastoiditis (4/9; 44%). Three guidelines (3/9; 33%) discussed imaging modalities such as a CT brain when investigating secondary mastoiditis.

### Approach to antibiotic administration
There were two approaches towards antibiotic administration: a watchful waiting approach and immediate antibiotic prescription (table 1). Fifteen of 17 (88%) of the European guidelines recommended a watchful waiting approach where clinicians were encouraged to prescribe antibiotics if symptoms persisted for 1–3 days or in case of any clinical deterioration. TM perforation/otorrhoea (14/15; 93%) and severity of symptoms (13/15; 87%) were the most common indications for immediate antibiotic administration (table 2). WHO guidelines recommended all children with confirmed AOM be given antibiotics.

### First-line antibiotic therapy
Of note, 14 of 17 (82%) European guidelines recommended oral amoxicillin as an option for first-line treatment (figure 3), of which 7/14 (50%) recommended a high dose (75–90 mg/kg/day) and 5/14 (36%) a low dose (30–60 mg/kg/day). Stratification to high-dose or low-dose amoxicillin for children in the UK SIGN guideline

**Table 2** Indications for consideration of immediate antibiotic treatment in European and AAP guidelines

| Guideline | Age (months)* | Parental input† | Unilateral AOM‡ | Bilateral AOM aged <24 months§ | Severe symptoms¶ | Co-morbidities | Recurrent AOM | TM perforation/ otorrhoea |
|---|---|---|---|---|---|---|---|---|
| Italy | - | - | + | + | + | - | - | + |
| Spain | <24 | - | - | + | + | - | + | + |
| Denmark | <6 | - | - | + | + | - | - | + |
| France | <24 | + | - | - | + | - | - | - |
| Portugal | <6 | - | - | + | + | - | + | + |
| USA | - | + | + | + | + | - | - | - |
| Norway | <12 | - | - | + | - | - | - | + |
| Poland | <6 | + | + | + | + | + | + | + |
| Belgium | <6 | - | - | + | + | + | - | + |
| Czech Republic | - | - | - | - | + | - | - | + |
| Finland | <24 | - | - | + | - | - | - | + |
| Germany | <24 | - | - | + | + | + | + | + |
| Ireland | - | - | - | - | - | - | - | + |
| Luxembourg | <24 | - | - | - | + | - | - | - |
| The Netherlands | <6 | - | - | + | + | + | - | + |
| Sweden | <12 | - | - | + | + | + | - | + |
| Switzerland | <24 | - | - | + | + | + | + | + |
| UK | - | - | - | - | - | - | - | - |

**Legend**

Highest grade

Moderate grade

No grade

'-' indicates that those indications are not mentioned in the guideline.

Note: There is no 'Lowest grade' in this table.

*Sweden: also children aged >12 years. Switzerland: <24 months of age, only if the child appears unwell.

†France: give antibiotics if parents are considered unreliable. USA: joint decision-making with parents at any age. Poland: joint decision-making with parents if child is <24 months of age.

‡Unilateral: Italy: if age <6 months. Poland: if age <24 months, then can give after joint decision-making with parents.

§Belgium, Finland and Sweden: bilateral at any age. Luxembourg: after consultation with parents. Switzerland: only if <24 months old.

¶Symptoms include fever, otalgia, pain, vomiting and diarrhoea.

AAP, American Association of Pediatrics; AOM, acute otitis media; TM, tympanic membrane.

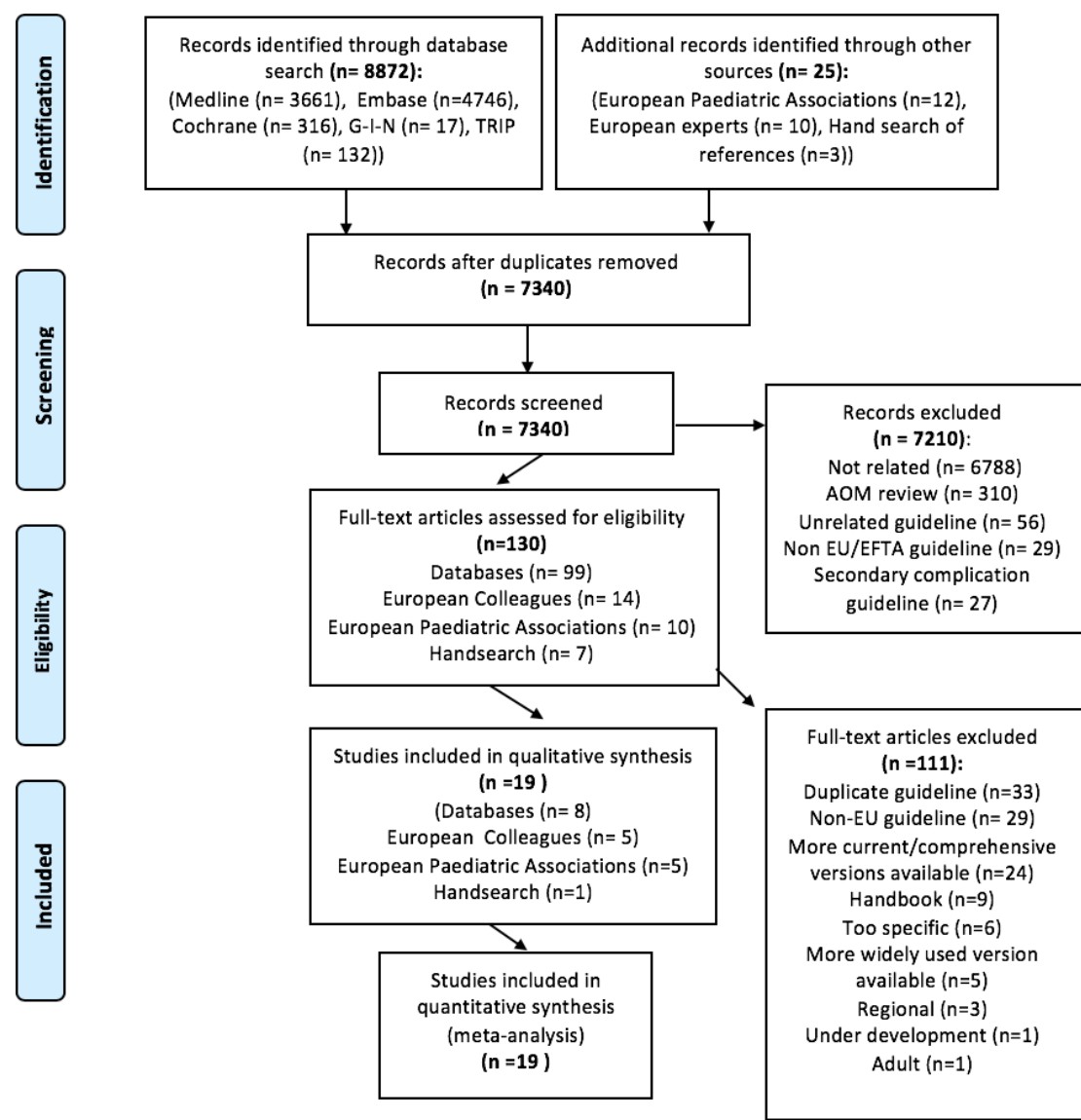

**Figure 1** PRISMA systematic review flow diagram

is weight-dependent; the Irish guidelines did not specify a dose. All the Nordic countries (ie, Denmark, Sweden and Norway) except Finland included oral penicillin V 24–75 mg/kg/day as a first-line choice (see online supplementary file 8).

### Treatment failure and penicillin allergy: alternative antibiotic treatments

In case of treatment failure, per oral/intravenous amoxicillin-clavulanic acid (11/15; 73%) and intravenous/intramuscular ceftriaxone (8/15; 53%) were the most commonly recommended antibiotics. In case of penicillin allergy, guidelines advised either oral clarithromycin (8/16; 50%) or oral trimethoprim–sulfamethoxazole (6/16; 38%) (see online supplementary file 8).

### Quality assessment: AGREE II scores

All guidelines were appraised using the AGREE II Criteria (table 3). In four of seven domains (ie, 2, 3, 5 and 6), European guidelines obtained a mean score of ≤41%

while only two domains (ie, 1 and 4) scored above 63% (see online supplementary file 9a,b)

### LoE and SoR

Of note, 10 of 17 European guidelines (59%) based their certainty of evidence (ie, LoE) and SoR on a variety of methodologies (table 4). The only crossover was between Poland and Spain which used a methodology from the Infectious Diseases Society of America. AGREE II scores for quality of the LoE were variable, and approximately half of European guidelines (8/17; 47%) scored ≤4 across all items. SoR was often based on study design (ie, multiple randomised controlled trials), but for some it was based on more subjective assessments (ie, 'well-conducted studies').

### Antibiotic stewardship

The majority of guidelines provided diagnostic criteria for AOM, specifications on when to start antibiotics, the route of administration and the duration of treatment (table 5).

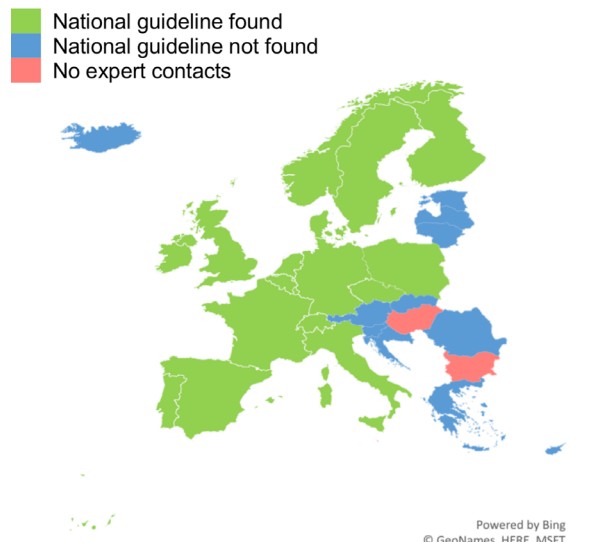

National guideline found
National guideline not found
No expert contacts

Powered by Bing
© GeoNames, HERE, MSFT

*National guidelines found:* Belgium (INAMI 2016), Czech Republic (CzMA 2011), Denmark (DSAM 2014), Finland (Duodecim 2017), France (AFSSAPS 2011), Germany (DEGAM 2014), Ireland (HSE 2012), Italy (SIP 2010), Luxembourg (CSDS 2007), Netherlands (NHG 2014), Norway (ASP 2016), Poland (NIL 2016), Portugal (DGS 2014), Spain (AEPED 2012), Sweden (MPA 2010), Switzerland (PIGS 2010), United Kingdom (SIGN 2003)

**Figure 2** European AOM guidelines (lead group and year published).

However, less than half referred to country-specific AMR patterns, and four (24%) included both country-specific AMR data and specified resistance levels to amoxicillin/amoxicillin–clavulanic acid to guide local choices.

## DISCUSSION

Approximately half of the 32 EU/EFTA countries have AOM guidelines. Diagnosis of AOM was based on similar signs and symptoms. Tympanocentesis was commonly reserved for treatment failure. The vast majority of European guidelines advocated for a watchful waiting approach to antibiotic therapy with the most common indications for treatment being TM perforation and severity of symptoms. Amoxicillin was the most commonly recommended first-line antibiotic but with differences in terms of

recommended duration and dosage. Our quality assessment found low mean AGREE II scores of ≤41% in most domains. Less than 60% of guidelines used a national or international system to rate LoE to support recommendations. Less than half of the guidelines referred to country-specific patterns of AMR.

Strengths of our study include the comprehensiveness of our three-pronged search strategy, the use of AGREE II, an internationally recognised guideline appraisal tool and an assessment of which LoE and SoR were used. Our analysis also included a qualitative assessment of whether antibiotic stewardship was considered in the development of guidelines based on five criteria. In order to provide a broad sense on whether AMR data were considered, one of the criteria was whether the antibiotic recommendations referred to national-level AMR data. However, the limitation of this is that there is often wide heterogeneity in AMR patterns within each country, therefore guidelines should ideally recommend that the antibiotic choice be adapted to available local AMR data. Another limitation is our focus on simple AOM and exclusion of guidelines about complex cases requiring otolaryngology specialist input.

Previously published works demonstrated a common consensus in criteria for AOM diagnosis, and that a watchful waiting period was the standard of care in Europe; amoxicillin was also found to be the most commonly recommended antibiotic.[43–45] In comparison with these studies, our work aimed to compare additional facets of AOM management in Europe, including grading their quality, comparison of LoE and SoR and assessing their inclusion of country-specific AMR data. Zeng *et al* also used AGREE II scores to assess quality of upper respiratory tract infections guidelines including three AOM guidelines from Japan, USA and UK.[46] We note a >10-point discrepancy in scoring in two of six domains between Zeng *et al* and ourselves for UK SIGN and US AAP AOM guidelines. This may indicate inter-user variability in AGREE II scoring.[47] Elias *et al* assessed global infectious diseases guidelines and found that local AMR

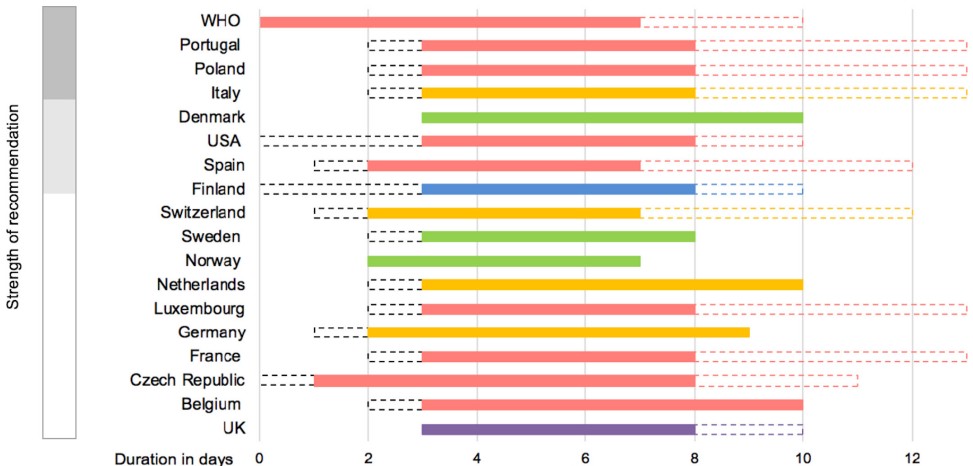

**Figure 3** Routine first-line antibiotics: initiation, choice, duration and Strength of Recommendation.

**Table 3** AGREE II scores (%) of European, AAP and WHO guidelines

| Domain number | Domain name | European mean (range) | AAP mean | WHO mean |
|---|---|---|---|---|
| 1 | Scope and purpose | 57 (10–100) | 97 | 94 |
| 2 | Stakeholder involvement | 41 (0–92) | 67 | 58 |
| 3 | Rigour of development | 34 (0–83) | 88 | 80 |
| 4 | Clarity of presentation | 78 (21–100) | 89 | 92 |
| 5 | Applicability | 23 (0–58) | 35 | 60 |
| 6 | Editorial independence | 29 (0–96) | 54 | 83 |

AAP, The American Association of Pediatrics; WHO, World Health Organisation.

patterns were taken into account in 50%–75% of recommendations which is similar to our findings.

The development of clinical guidelines according to the high standards of the AGREE II criteria is a resource-intensive exercise and this may be one of the reasons why we did not identify any guidelines from some countries. Many guidelines in this study received low AGREE II scores. Many of the resource-intensive initial steps in guidelines development are universal, for example defining the objectives, the clinical questions, the target populations of patients and end users and designing a comprehensive search strategy to identify relevant evidence from the literature, a process to appraise the evidence, a way to present recommendations unambiguously and strategies to successfully implement guidelines. Replicating this process in each country to reach similar conclusions does

**Table 4** Level of Evidence in AOM guidelines

| Country | Grading system for LoE * | Score: consideration of benefits and harms (AGREE II Item 11†) | Score: strengths and limitations of the evidence (AGREE II Item 9) | Score: link between recommendations and evidence (AGREE II Item 12) |
|---|---|---|---|---|
| Belgium | INAMI | 5 | 7 | 6 |
| Czech Republic | – | 1 | 1 | 2 |
| Denmark | OCEBM | 7 | 7 | 6 |
| Finland | Duodecim | 1 | 6 | 6 |
| France | ANAES | 3 | 1 | 1 |
| Germany | AWMF | 6 | 3 | 3 |
| Ireland | – | 1 | 1 | 1 |
| Italy | PNLG | 5 | 5 | 6 |
| Luxembourg | – | 3 | 1 | 2 |
| The Netherlands | – | 7 | 7 | 5 |
| Norway | – | 1 | 1 | 3 |
| Poland | Infectious Disease Society of America | 6 | 3 | 5 |
| Portugal | European Society of Cardiology | 2 | 2 | 4 |
| Sweden | – | 3 | 3 | 1 |
| Switzerland | – | 1 | 1 | 1 |
| Spain | Infectious Disease Society of America | 5 | 2 | 7 |
| UK | SIGN | 7 | 7 | 6 |
| USA | AAP | 7 | 7 | 7 |
| WHO | GRADE | 7 | 7 | 6 |

*If no LoE scale used, it is denoted by –.
†AGREE II scores: 1=no information in the guideline; 7=exceptional reporting.
AAP, The American Association of Pediatrics; ANAES, l'Agence Nationale d'Accréditation et d'Évaluation en Santé ; AOM, acute otitis media; AWMF, Arbeitsgemeinschaft der Wissenschaftlichen Medizinischen Fachgesellschaften; GRADE, Grading of Recommendations, Assessment, Development and Evaluations; INAMI, Institut National d'Assurance Maladie-Invalidité; LoE, level of evidence; OCEBM, Oxford Centre for Evidence Based Medicine; PNLG, Programma Nazionale Linee Guida; SIGN, Scottish Intercollegiate Guidelines Network.

**Table 5** Antibiotic stewardship and AOM guidelines

| | Do guidelines provide diagnostic criteria? | Do guidelines specify when to initiate antibiotics? | Do guidelines specify route of administration? | Do guidelines specify duration of antibiotic regimens? | Do antibiotic recommendations refer to country-specific AMR patterns? | | |
| --- | --- | --- | --- | --- | --- | --- | --- |
| | | | | | Percentage of antibiotic recommendations that refer to country-specific AMR patterns (%) | Amoxicillin dosage that refers to country-specific AMR patterns | Amoxicillin–clavulanic acid dosage that refers to country-specific AMR patterns |
| Belgium | Yes | Yes | Yes | Yes | 80 | Yes | Yes |
| Czech Republic | Yes | Yes | Yes | Yes | 0 | Unclear | Unclear |
| Denmark | Yes | Yes | Yes | Yes | 0 | Not applicable | Unclear |
| Finland | Yes | Yes | Yes | Yes | 63 | Yes | Yes |
| France | Yes | Yes | Yes | Yes | 0 | Unclear | Unclear |
| Germany | Yes | Yes | Yes | Yes | 0 | Unclear | Unclear |
| Ireland | Unclear | Yes | Yes | Unclear | 0 | Unclear | Not applicable |
| Italy | Yes | Yes | Yes | Yes | 67 | Yes | Yes |
| Luxembourg | Yes | Yes | Yes | Yes | 0 | Unclear | Unclear |
| The Netherlands | Yes | Yes | Yes | Yes | 100 | Yes | Yes |
| Norway | Yes | Yes | Yes | Yes | 0 | Unclear | Not applicable |
| Poland | Yes | Yes | Yes | Yes | 100 | Yes | Not applicable |
| Portugal | Yes | Yes | Yes | Yes | 71 | Yes | Yes |
| Spain | Yes | Yes | Yes | Yes | 100 | Yes | Yes |
| Sweden | Yes | Yes | Yes | Yes | 100 | Unclear | Not applicable |
| Switzerland | Unclear | Yes | Yes | Yes | 0 | Unclear | Unclear |
| UK | Yes | Yes | Yes | Yes | 0 | Unclear | Unclear |
| USA | Yes | Yes | Yes | Yes | 100 | Yes | Yes |
| WHO | Yes | Yes | Yes | Yes | Not applicable | Not applicable | Not applicable |

**Legend**

Promotes antibiotic stewardship

Partially promotes antibiotic stewardship

Does not promote antibiotic stewardship

AMR, antimicrobial resistance; AOM, acute otitis media.

not seem necessary nor efficient, and it may make sense for these or some of these processes to be undertaken by a core group of experts from across Europe. This is already the case for other medical specialities, for example the European Joint Task Force for cardiovascular disease prevention provides guidelines that can be used across Europe.[48] The centrally developed guidelines could then be adapted in each country for recommendations, such as choice of antibiotics, which depends on local AMR patterns and immunisation coverages against the main pathogens causing AOM. This implies the implementation of robust epidemiological and standardised AMR surveillance systems in each country which is currently underway with the support of international initiatives such as the European Centre for Disease Prevention and Control surveillance systems,[49] and the WHO Global Antimicrobial Resistance Surveillance System.[50] Other aspects that could lead to local adaptation could be local care pathways, and user and patient preferences. This approach would allow the development of guidelines of better quality and better adapted to local contexts, and it might contribute to reducing the spread of AMR.

## CONCLUSION

Review of guidelines reveals major similarities in AOM management recommendations across Europe. Existing European guidelines scored poorly in most AGREE II domains, including items related to how evidence was gathered and appraised. Consideration of country-specific antibiotic resistance patterns appears to be limited. Centrally produced guidelines adapted for local care pathways, user and patient preferences, and for local antimicrobial resistance patterns may provide more targeted recommendations, reduce unnecessary antibiotic administration and help reduce the spread of antibiotic resistance.

**Acknowledgements** We would like to acknowledge Professor Mike Levin and our colleagues from the PERFORM consortium, Young ESPID, and all our contacts who helped us in obtaining national guidelines: Daniela Klobassa (Austria), Jan Verbakel (Belgium), Žaneta Jelčić (Croatia), Stephanie Menikou (Cyprus), Linda Alderson (Czech Republic), Ivan Peychl (Czech Republic), Anna Turkova (Czech Republic), Rikke Jorgensen (Denmark), Juri Lindy Pedersen (Denmark), Inga Ivaskeviciene (Estonia), Piia Jogi (Estonia), Irja Lutsar (Estonia), Eda Tamm (Estonia), Esa Korpi (Finland), Niina Valtanen (Finland), Romain Basmaci (France), Jean Christophe Mercier (France), Christoph Bidlingmaier (Germany), Ulrich von Both (Germany), Florian Gothe (Germany), Johanna Krone (Germany), Konstatinos Kakleas (Greece), Valtyr Thor (Iceland), Maeve Kelleher (Ireland), Silvia Bressan (Italy), Dace Zavadska (Latvia), Simona Sabulyte (Lithuania), Armand Biver (Luxembourg), Simon Attard-Montalto (Malta), David Pace (Malta), Ingebjørg Fagerli (Norway), Arne Martin Slåtsve (Norway), Magdalena Marczyńska (Poland), Maria Pokorska-Śpiewak (Poland), Kacper Toczylowski (Poland), Carolina Costa (Portugal), Irina Branescu (Romania), Diana Moldovan (Romania), Laszlo Kovacs (Slovakia), Keti Vincek (Slovenia), Pablo Obando Pacheco (Spain), Irina Rivero (Spain), Frederico Maritinon Torres (Spain), Giannos Orfanos (Sweden), Philipp Agyeman (Switzerland), Hermione Lyall (UK), and Elizabeth Whittaker (UK).

**Contributors** SY conceived the study. HS, JED, RN and SY all contributed to the study design. HS was responsible for the systematic database search. JED, RN and SY all contacted experts in their scientific networks to obtain additional guidelines and check the use and validity of those identified. HS and JED were responsible for data extraction including LoE, SoR and antibiotic stewardship and AGREE II scoring. HS, JED, RN and SY all contributed to the interpretation of the results, the drafting and revision of the manuscript and they agree with the final version.

**Funding** RN was supported by NIHR Academic clinical fellowship and lectureship award programme. JED and SY are supported by PERFORM, a consortium funded by the European Union's Horizon 2020 programme, under grant agreement No. 668303.

**Disclaimer** The funding sources did not take part in the design, analysis, interpretation of data, writing of the report or decision to submit the article for publication. All authors had full access to all the data, and they can take responsibility for the integrity of the data and the accuracy of the data analysis.

**Map disclaimer** The depiction of boundaries on the map(s) in this article do not imply the expression of any opinion whatsoever on the part of BMJ (or any member of its group) concerning the legal status of any country, territory, jurisdiction or area or of its authorities. The map(s) are provided without any warranty of any kind, either express or implied.

**Competing interests** None declared.

**Patient consent for publication** Not required.

**Ethics approval** None.

**Provenance and peer review** Not commissioned; externally peer reviewed.

**Data availability statement** Data are available upon reasonable request. The primary data for this study was treatment guidelines, and these can be shared on request to the corresponding author.

**ORCID iDs**
Hijiri G Suzuki http://orcid.org/0000-0003-3134-0297
Juan Emmanuel Dewez http://orcid.org/0000-0002-0323-6217
Ruud G Nijman http://orcid.org/0000-0001-9671-8161
Shunmay Yeung http://orcid.org/0000-0002-0997-0850

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
