## [Reviewer comments · BMJ Open]

ARTICLE DETAILS

TITLE (PROVISIONAL)	Clinical practice guidelines for acute otitis media in children: A systematic review and appraisal of European national guidelines
AUTHORS	Suzuki, Hijiri; Dewez, Juan; Nijman, R; Yeung, Shunmay

VERSION 1 - REVIEW

REVIEWER	Richard M. Rosenfeld, MD, MPH, MBA Department of Otolaryngology, SUNY Downstate Medical Center, Brooklyn, NY, USA None financial. Potential intellectual conflict as an author of the AAP guideline on acute otitis media.
REVIEW RETURNED	10-Nov-2019

GENERAL COMMENTS	The authors conduct a high-quality systematic review of European guidelines for managing acute otitis media (AOM) in children. Given the diversity of guidelines that exist for AOM, in terms of methodologic rigor and breadth of conclusions, a systematic review could provide valuable information for harmonizing approaches and guiding future efforts. General comment: The guidelines from the USA, American Academy of Pediatrics (AAP) are incorrectly referred to as the American Association of Paediatrics. I appreciate that these are included, but why they are present in a review of European guidelines is unclear. My impression, as one of the authors of the 2013 AAP guidelines, is that many European, and other, countries have replicated or adapted (very closely) many of the recommendations in their own guidelines. Since the AAP guidelines appear to be a standard (or template) on which others are based, perhaps indicating concordance or discordance with the major recommendations in this guideline (e.g., diagnostic criteria, indications for watchful waiting, definition of treatment failure) would be useful. This is done in some of the tables (along with the WHO guideline) but would benefit from more explicit explanation up front in the manuscript and consistency with the comparisons within. Introduction: The authors state (In 16) “approximately half of all patient with AOM get better without treatment,” which presumably intends to frame the discussion regarding natural history and limited antibiotic impact. Of note, the 50% rate is deceptive because it is based on 2 randomized trials (references 9 and 10) that both included abnormal TM appearance on otoscopy as a criterion for treatment “failure,” even when the children were
--

	symptomatically improved or resolved. Most systematic reviews, however, show about 80% spontaneous symptom resolution in placebo groups and the number needed to treat for benefit with antibiotic (compared to placebo) is 20 (reference #13, Cochrane review, which should be updated to 2015 revision). Antibiotic stewardship: Using a scoring system for this is a nice approach, but I would question the validity of a question regarding “local resistance patterns.” Resistance patterns are indeed local, often varying greatly in proximate communities or even with hospitals or other healthcare facilities. At the national level, where most of these guidelines were developed, “local” resistance patterns would not be meaningful because of heterogeneity in smaller communities. The role of local resistance patterns is for the guideline end-user to consider them regarding choice of antibiotic. This may explain the poor results regarding linkage of antibiotic recommendations to local resistance patterns seen in Table 4. Strength of recommendation: There is an inherent flaw in basing Strength of recommendation (SoR) on the level of evidence in the OCEBM table. SoR in guideline development is also dependent on the relative balance of benefit vs. harm in following a recommendation, based on the underlying confidence in the quality, consistency, and directness of the evidence. There should be 2 separate processes: one for determining a level of aggregate evidence and another for rating confidence of evidence and benefit vs. harm balance. Rather than simply report SoR based on and OCEBM conversion, it would be more meaningful to know if the guideline developers (a) rated aggregate level of evidence (and if so, using what scale/criteria), (b) did an independent balance vs. harm/risk assessment, (c) explicitly considered the confidence in the level of evidence (and upgraded, or downgraded accordingly), and (d) formulated a SoR based on (a) and (b) (or some other stated methodology). Table 2, pg. 10: This is titled “Indications for consideration of immediate antibiotic treatment in European and AAP guidelines,” but the title is deceptive. For example, the AAP guidelines make clear that any case of AOM with a certain diagnosis (based on a distinctly bulging TM) could be considered for immediate antibiotics, and that it would not be inappropriate to prescribe antibiotics in that circumstance. For many cases, however, it would also be appropriate to consider or recommend watchful waiting with a deferred antibiotic prescribing strategy. The way the table is currently presented would suggest that unilateral AOM would not be considered for immediate antibiotics in the AAP guideline, which is an incorrect inference.
--	--

REVIEWER	Tal Marom Head of Pediatric Otolaryngology Unit, Department of Otolaryngology-Head and Neck Surgery Assuta Ashdod University Hospital 7747629 Ashdod Israel
REVIEW RETURNED	12-Nov-2019

GENERAL COMMENTS	Thank you for this interesting work. Well written and definitely deserved publication. Minor issues: do we really/can we really structure central European guidelines for AOM treatment at times when PCVs are not available in all countries, and at times when we do not have bacterial resistance data from AOM in all countries? not sure. Introduction. Please state that AOM can be a viral disease and bacterial super-infection is considered to be a complication; this is the reason why not all AOM episodes should be treated with antibiotics. Do you have references for antibiotic resistance specifically from AOM cultures? if yes, please provide some data. Your major limitation is that you included only guidelines from Central and Western Europe and from Eastern Europe.. why do you think this is the reason and what could be done in order to improve the suggested surveillance?
--

VERSION 1 – AUTHOR RESPONSE

Reviewers feedback	Our response
Reviewer 1	
The guidelines from the USA, American Academy of Pediatrics (AAP) are incorrectly referred to as the American Association of Paediatrics.	We thank the reviewer for highlighting this and we have amended the manuscript accordingly.
I appreciate that these are included, but why they are present in a review of European guidelines is unclear.	Thank you for your query. We have included the AAP and WHO guidelines for comparison as we feel these are both widely used and recognised by our European and international readers. This was strengthened by our findings summarised in Table 1 included in this document (see below), whereby some national guidelines explicitly state they have based their recommendations upon AAP guidelines. We have added this explanation to lines 251-253 of Methods section.
My impression, as one of the authors of the 2013 AAP guidelines, is that many European, and other, countries have replicated or adapted (very closely) many of the recommendations in their own guidelines. Since the AAP guidelines appear to be a standard (or template) on which others are based, perhaps indicating concordance or discordance with the major recommendations in this guideline (e.g., diagnostic criteria, indications for	We appreciate the Reviewer’s comments. We have reviewed the guidelines to identify if they state that the AAP recommendations were used as a basis for their guidelines (Table 1 of this rebuttal letter, see below). Most guidelines (12/17) did not clearly state that the guidelines had been based on the AAP guidelines specifically. Those that do also included other national guidelines as a basis for comparison, which has been clarified in lines 335-340 Therefore, we felt it would be beneficial to keep our approach to compare European guidelines, and include the AAP and WHO as important international reference points.

watchful waiting, definition of treatment failure) would be useful. This is done in some of the tables (along with the WHO guideline) but would benefit from more explicit explanation up front in the manuscript and consistency with the comparisons within.

Table 1: Are European AOM recommendations explicitly based upon other guidelines?	
Country	Does the guideline clearly state that it is based on the AAP guideline?
Belgium	No
Czech Republic	No
Denmark	No
Finland	No
France	No
Germany	Includes comparison of several guidelines from different countries, including the AAP
Ireland	No
Italy	Yes- AAP (2004 guideline) (Page 4) In addition to guideline of the American Academy of Pediatrics (AAP) and American Academy of Family Physicians, traditionally regarded as a point of excellent reference for the Italian pediatrician, published in 2004 (AAP 2004), I have also published a number of LG OMA, designed to meet national or regional needs of individual professional organizations.
Luxembourg	Yes- AAP (2004 guideline) and French AFSSAPS (Page 1) Recommendations for the diagnosis and treatment of acute otitis media have been drawn up on the basis of the recent French
	recommendations (Http://www.afssaps.sante.fr/) and American (American Academy of Pediatrics 2004, http://pediatrics.aappublications.org/cgi/content/full/114/2/S2/555) (Page 1)
Netherlands	No
Norway	No

	Poland	To some extent – AAP, SIGN. A very important guidepost in developing guidelines for infections of the upper and lower respiratory tract was the creation in 2001 of the European working group, with the task of developing recommendations for the treatment prescription drugs in respiratory infections, and the number of subsequent recommendations of both the European and American [6-10]. (Page 66) Clinical diagnosis Proper identification has a crucial role in proposing appropriate therapy as well as in evaluating the effect on the course of the disease. AOM definitions are adopted from the American Academy of Pediatrics and the Scottish Intercollegiate Guidelines Network SIGN [40, 41], although often criticized for insufficiently precise to differentiate AOM with exudative ear infections, led to strengthening the criteria for diagnosis In both studies, the diagnosis of AOM was determined based on the criteria developed by the American Academy of Pediatrics and compared the effects of treatment with amoxicillin-clavulanic acid to placebo.
	Spain	No
	Sweden	Includes comparison of several guidelines from different countries, including the AAP
	Switzerland	No

	United Kingdom	No
	WHO	No

Introduction: The authors state (In 16) “approximately half of all patient with AOM get better without treatment,” which presumably intends to frame the discussion regarding natural history and limited antibiotic impact. Of note, the 50% rate is deceptive because it is based on 2 randomized trials (references 9 and 10) that both included abnormal TM appearance on otoscopy as a criterion for treatment “failure,” even when the children were symptomatically improved or resolved. Most systematic reviews, however, showw about 80% spontaneous symptom resolution in placebo groups and the number needed to treat for benefit with antibiotic (compared to placebo) is 20 (reference #13, Cochrane review, which should be updated to 2015 revision).	We appreciate the Reviewer’s comments and we welcome the suggested reference. We have included in our Introduction section Line 219-220. We have updated the Cochrane review reference to the 2015 revision, line 798.
Antibiotic stewardship: Using a scoring system for this is a nice approach, but I would question the validity of a question regarding “local resistance patterns.” Resistance patterns are indeed local, often varying greatly in proximate communities or even with hospitals or other healthcare	Thank you for the generally positive reception to our approach. We agree that the AMR patterns mentioned in Table 4 (labelled Table 5 in the revised manuscript) of the manuscript are actually national AMR patterns rather than “local” and have changed the wording to “country-specific” in the table and in the text, lines 517-518. Our aim was to get a sense on whether recommendations were based on any type of AMR data. We believe that national guidelines should ideally provide both a general recommendation for antibiotic choice and dose referring to the microbiological data on which the recommendation is based, and advise that the antibiotic choice (and dose) should
facilities. At the national level, where most of these guidelines were developed, “local” resistance patterns would not be meaningful because of heterogeneity in smaller communities. The role of local resistance patterns is for the guideline end-user to consider them regarding choice of antibiotic. This may explain the poor results regarding linkage of antibiotic recommendations to local resistance patterns seen in Table 4.	be modified based on local AMR data if available. We have clarified this in the Discussion and added a note about the limitations of this indicator (Lines 555-563). The countries highlighted in green in Table 4 (labelled Table 5 in the revised manuscript), Column 4, are countries which guidelines provided references to AMR data from the country, to support their recommendations.

Strength of recommendation: There is an inherent flaw in basing Strength of recommendation (SoR) on the level of evidence in the OCEBM table. SoR in guideline development is also dependent on the relative balance of benefit vs. harm in following a recommendation, based on the underlying confidence in the quality, consistency, and directness of the evidence. There should be 2 separate processes: one for determining a level of aggregate evidence and another for rating confidence of evidence and benefit vs. harm balance. Rather than simply report SoR based on and OCEBM conversion, it would be more meaningful to know if the guideline developers (a) rated aggregate level of evidence (and if so, using what scale/criteria), (b) did an independent balance vs. harm/risk assessment, (c) explicitly considered the confidence in	Thank you for the insightful comment. We fully agree with the Reviewer that there are inherent flaws in using Strength of recommendation (SoR). We had initially attempted to compare LoE between national guidelines compared with OCEBM table. Unfortunately, due to the heterogeneity between the national Level of Evidence (LoE) utilised, we were unable to make any meaningful comparisons (Supplementary File 3-4). We agree that it is beneficial to add more meaningful information about the LoE included. The framework suggested by the Reviewer allows providing information about the LoE and SoR that were used by the guideline developers. We have therefore revisited the guidelines and collected supplementary data We have slightly modified the points as suggested by the Reviewer: a) Rated aggregate level of evidence: Yes/No. If yes, what scale did they use: Name of scale b) Independent harm vs risk assessment: We felt this was very close to Item 11 on the AGREE II Instrument (“The health benefits, side effects, and risks have been considered in formulating the recommendations”) and therefore have included the score as a way to be able to objectively compare across guidelines. The possible scores range from 1 (no information) to 7 (exceptional reporting).
the level of evidence (and upgraded, or downgraded accordingly), and (d) formulated a SoR based on (a) and (b) (or some other stated methodology).	c) Explicitly considered the confidence in the level of evidence: We felt this was very close to Item 9 on the AGREE II Instrument (“The strengths and limitations of the body of evidence are clearly described”) and therefore have included the score. d) formulated a SoR: Again, we felt this was very similar to Item 12 on the AGREE II Instrument (“There is an explicit link between the recommendations and the supporting evidence.”) and therefore have included the score given. This has been included as a new Table 4: Level of evidence in AOM guidelines (under line 511). We would also be in agreeance for this table to be shifted into the Appendix if this was felt to be more appropriate in what we believe is an already fairly detailed description of multiple aspects of AOM management. We would be interested to know the Reviewer’s opinion about this.

	We have included this in lines 498-503 a summary of the results in Table 4.
Table 2, pg. 10: This is titled “Indications for consideration of immediate antibiotic treatment in European and AAP guidelines,” but the title is deceptive. For example, the AAP guidelines make clear that any case of AOM with a certain diagnosis (based on a distinctly bulging TM) could be considered for immediate antibiotics, and that it would not be inappropriate to prescribe antibiotics in that circumstance. For many cases, however, it would also be appropriate to consider or recommend watchful waiting with a deferred antibiotic prescribing strategy. The way the table is currently presented would suggest that unilateral AOM would not be considered for immediate antibiotics in the AAP guideline, which is an incorrect inference.	Thank you to the Reviewer for highlighting that the headings of Table 2 inadvertently inferred unilateral AOM treatment would not be an indication for immediate treatment. We have therefore included a column titled “Unilateral AOM” to Table 2 under line 421. To present the recommendations with more precision, we have added a new row to Table 1, under line 383 titled “Immediate antibiotics for any AOM can be considered.” To clarify, we have added “WHO guidelines recommended all children with confirmed AOM be given antibiotics” in lines 377-378. We have also updated Figure 3 accordingly.
Reviewer 2	
Minor issues: do we really/can we really structure central European guidelines for AOM treatment at times when PCVs are not available in all countries, and at times when we do not have bacterial resistance data from AOM in all countries? not sure	We agree that some recommendations (such as the decision to start antibiotics immediately, or the choice of antibiotics) will definitely need to be adapted locally, depending on AMR patterns and the immunisation coverage against the main pathogens causing AOM. However, we felt that centralising the processes that would be common for all organisation developing guidelines (such as defining the guidelines’ objectives, the relevant clinical questions, the target population of patients and guidelines users, a standardised search strategy to access evidence, the vaccination coverage thresholds upon which initiation of antibiotics could be delayed, etc) would allow avoiding duplication of efforts and saving resources, as not all countries have dedicated and funded bodies (such as the National Institute for Health and Care Excellence in the UK) to perform these tasks. We have amended the text in the discussion to make it clearer (lines 596-605), and lines 633-634 and provide more detailed about which processes could be made centrally or locally.

Please state that AOM can be a viral disease and bacterial super-infection is considered to be a complication; this is the reason why not all AOM episodes should be treated with antibiotics.	We agree with the comment and amended the introduction section accordingly in lines 208209
Do you have references for antibiotic resistance specifically from AOM cultures? if yes, please provide some data.	Some of the guidelines broadly discuss antimicrobial resistance patterns of the bacteria causing AOM. Unfortunately, we did not identify any specific information within guidelines about if country-specific antimicrobial resistance data was from cultures obtained from middle ear cultures (the ideal scenario) or from broader upper respiratory tract cultures.
Your major limitation is that you included only guidelines from Central and Western Europe and from Eastern Europe..	We haven't included other guidelines from outside Europe because that was out of our scope and resources. We agree that it would be important to do it, and we are keen to share our methods and results with any researcher interested in repeating this for countries outside Europe. In terms of Eastern European countries, we have included most European countries in our comprehensive search of guidelines, but we haven't identified guidelines from Eastern Europe. Our collaborators from those countries confirmed there were no national guideline in their countries.
why do you think this is the reason and what could be done in order to improve the suggested surveillance?	We think it might be because it is too resource intensive to develop guidelines, particularly if other, not very distant, countries have developed guidelines that are publicly available. We have clarified this in lines 591-594. Including representatives of those countries in a centralised development process could be a way to ensure views from across Europe are considered. We added this consideration in lines 605-608). In terms of improving the local surveillance for AMR, we suggest that countries join international efforts such as the European Centre for Disease Prevention and Control (ECDC) surveillance systems, or the WHO Global Antimicrobial Resistance Surveillance System (GLASS), which already provide methodological support to participating countries and gather data from several countries. We have added these considerations in the discussion section (Lines 626-632).